# Antagonistic Effect of *Trichoderma longibrachiatum* (TL6 and TL13) on *Fusarium solani* and *Fusarium avenaceum* Causing Root Rot on Snow Pea Plants

**DOI:** 10.3390/jof8111148

**Published:** 2022-10-29

**Authors:** Thomas Afriyie Boakye, Huixia Li, Richard Osei, Solomon Boamah, Zhang Min, Chunhui Ni, Jin Wu, Mingming Shi, Wanqiang Qiao

**Affiliations:** 1College of Plant Protection, Gansu Agricultural University, Lanzhou 730070, China; 2Biocontrol Engineering Laboratory of Crop Diseases and Pests of Gansu Province, Lanzhou 730070, China

**Keywords:** root rot, snow pea, *Trichoderma*, *Fusarium*, biological control agent, antagonistic, antioxidant activities

## Abstract

Snow pea root rot in China is caused by *Fusarium solani* (FSH) and *Fusarium avenaceum* (FAH), which affect snow pea production. The chemical control methods used against FSH and FAH are toxic to the environment and resistance may be developed in persistence applications. Therefore, an alternative approach is needed to control these pathogens. This study focuses on *Trichoderma longibrachiatum* strains (TL6 and TL13), mycoparasitic mechanisms of FSH and FAH, as well as growth-promoting potentials on snow pea seedlings under FSH and FAH stress at the physiological, biochemical, and molecular levels. The average inhibitory rates of TL6 against FSH and FAH were 54.58% and 69.16%, respectively, on day 7. Similarly, TL13 average inhibitory rates against FSH and FAH were 59.06% and 71.27%, respectively, on day 7. The combined TL13 and TL6 with FSH and FAH reduced disease severity by 86.6, 81.6, 57.60, and 60.90%, respectively, in comparison to the controls. The snow pea plants inoculated with FSH and FAH without TL6 and TL13 increased malondialdehyde (MDA) and hydrogen peroxide (H_2_O_2_) contents in the leaves by 64.8, 66.0, 64.4 and 65.9%, respectively, compared to the control. However, the combined FSH and FAH with TL6 and TL13 decreased the MDA and H_2_O_2_ content by 75.6, 76.8, 70.0, and 76.4%, respectively, in comparison to the controls. In addition, the combined TL6 + FSH and TL6 + FAH increased the activity of superoxide dismutase (SOD), peroxidase (POD), and catalase (CAT) by 60.5, 64.7, and 60.3%, respectively, and 60.0, 64.9, and 56.6%, respectively, compared to the controls. Again, compared to the controls, the combined TL13 + FSH and TL13 + FAH increased the activity of SOD, POD, and CAT by 69.7, 68.6, and 65.6%, respectively, and 70.10, 69.5, and 65.8%, respectively. Our results suggest that the pretreatment of snow pea seeds with TL6 and TL13 increases snow pea seedling growth, controls FSH and FAH root rot, increases antioxidant enzyme activity, and activates plant defense mechanisms. The TL13 strain had the greatest performance in terms of pathogen inhibition and snow pea growth promotion compared to the TL6 strain.

## 1. Introduction

Snow peas (*Pisum sativum* L. var. *saccharatum*), also known as small cold beans [1], are edible, rich in nine kinds of amino acids needed by the human body, and have a high nutritional value [2]. Currently, China’s snow pea cultivation area and output account for 57.82% and 60.53%, respectively, of the world’s total production, second only to Canada [3], and occupy a pivotal position in the world’s snow pea production. At the same time, snow peas have a high yield and a high output value, and they have also become the main economic crop in many areas [4]. However, with the continuous expansion of the planting area of snow peas, the problems of various diseases have become increasingly prominent, so the prevention and control of snow pea diseases are very imminent.

*Fusarium* spp., *Pythium* spp., and *Rhizoctonia solani* are fungi that cause the soil-borne disease known as “root rot” [5]. It is prevalent in every pea-growing region in the world. *Fusarium* spp. is a soil-borne pathogen noted for growing on cereal crops, such as grain, straw, and hay, while several species can rarely be identified on various substrates. In many pea-growing regions across the world, root rot caused by *F. solani* and *F. avenaceum* is a significant challenge, resulting in massive crop losses [6]. Pathogens are considered to highly persist in soil and are capable of living in contaminated environments for relatively long periods, making management very challenging. *Fusarium* root rot is identified by the presence of reddish-brown lesions, as well as by tap roots and lower hypocotyls [7]. Plant-diseased regions expand and eventually turn brown as they mature. In older lesions, longitudinal cracks may occur, and cortical tissues may become discolored and decay. Under water stress conditions, root rots are very severe [8]. It occurs in the seedling stage and mainly damages the stem base and root system. *Fusarium*-infected stems have been found to have high levels of toxin accumulation. *Fusarium* infestation normally develops at the seedling stage and may even persist until the stage of maturity, so the infectious period is greater compared to other soil-borne infections. Root rot has a significant influence on both the quality and productivity of snow peas [9], and it occurs all year round in various production areas. *Fusarium* infections, particularly those that are soil-borne due to their mode of spread or dispersion, significantly impede the production of food plants (such as the snow pea) and are highly challenging to control.

Stress is a well-known cause of the accumulation of reactive oxygen species (ROS) [10]. Antioxidant system activation is a significant response approach for reducing ROS generation and decreasing oxidative damage under stress conditions. To effectively minimize ROS damage, plants have evolved a scavenging mechanism that comprises enzymatic and non-enzymatic antioxidants. Plant stress tolerance is primarily measured by these indicators: superoxide dismutase (SOD); peroxidase (POD); and catalase (CAT). These ROS scavenging activities, which are controlled by antioxidative enzymes, are the plant’s initial defense mechanism against stress, and they instantly exhibit the impacts of stressful conditions [11]. An effective antioxidant capacity is required to keep the equilibrium between the production of ROS and absorption, and to attenuate the negative impacts of stress on plant metabolism and development [10]. The reactions of plants to combined biotic and abiotic stresses are complicated by the complex interaction among plants, pathogens, and abiotic factors. Plant physiological and molecular responses to combined biotic pathogen stresses vary considerably from individual stresses [12,13]. Reactive oxygen species (ROS) induce chlorophyll breakdown and membrane lipid peroxidation, decreasing membrane permeability and sensitivity [14]. MDA is a product of unsaturated fatty acid peroxidation in phospholipids that causes cell membrane damage [15]. Chlorophyll depletion and lipid peroxidation are measured by malondialdehyde content (MDA) [16]. Plants evolve a variety of enzymatic (e.g., catalase, glutathione reductase, and several peroxidases) and non-enzymatic (ascorbate, carotenoids, flavonoids, and other phenolic compounds, etc.) detoxification systems to alleviate ROS and protect cells from oxidative damage [17]. Carotenoids are pigments that play several roles in plants, including direct photosynthesis and oxidative stress defense mechanisms [18]. The overproduction of ROS is a typical event in plants under both biotic and abiotic stress conditions, and it can induce oxidative damage to macromolecules and cell structures, resulting in plant growth and development inhibition, or even death. Among all ROS, H_2_O_2_ is a key factor in stress-signal-transduction pathways related to abiotic and biotic stress tolerance [19]. The overproduction of ROS is a typical event in plants under both biotic and abiotic stress conditions, and it can induce oxidative damage to macromolecules and cell structures, resulting in plant growth and development inhibition, or even death. Low levels of MDA and H_2_O_2_ in plants can minimize the negative impacts of stress by stimulating antioxidant systems and reducing ROS production. At the same time, this mechanism is significant to the improvement of agricultural productivity in stressed environments. Although numerous studies report strategies to enhance biotic and abiotic stress tolerance, there is no information available on the chlorophyll and carotenoid content, MDA, H_2_O_2_, antioxidant enzyme activities, and physiological mechanisms of stress reduction in plants specific to snow pea.

*Trichoderma* spp. are a well-known species of biocontrol fungi that are primarily isolated from soil, making them ideal for managing soil-borne diseases, such as root rot. *Trichoderma* species have been reported as a genus of soil-dwelling, teleomorphic filamentous fungi of rhizosphere microorganisms to enhance their host’s nutritional condition [20,21], provide modifications in the host’s physiology, and exude from roots [22]. For decades, *Trichoderma* spp. have been employed as biocontrol agents to manage a variety of plant fungal infections [23]. *Trichoderma* has been shown to reduce the occurrence of diseases, as well as a number of propagules of various soil-borne diseases, including *Fusarium* [24,25]. It serves as an efficient biological control agent in agriculture. It also promotes plant development and enhances the fertility of the soil due to its disease-reduction and composting abilities [26]. It is generally accepted that *Trichoderma’s* management of root rot is governed by numerous mechanisms. The ability of *Trichoderma* to manage soil-borne plant diseases via mycoparasitism, as well as antibiosis, is well reported [27]. It effectively promotes plant development, owing to its efficiency in the rhizosphere [28], capacity to produce or stimulate hormone production in plants [29], the release of available nutrients from the soil, and the enhancement of root-system-modification development [30]. *Trichoderma* enhances plant disease resistance by developing systemic resistance [31]. *Trichoderma*’s resistance to various biotic and abiotic stresses makes it a preferred choice for the control of integrated plant pathogens utilizing physical, chemical, and biological techniques [32]. *Trichoderma* generates and releases a wide range of chemicals that can cause localized or systemic defense mechanisms that help in protecting plants from the seedling stage to the maturity stage [33]. *T. harzianum* was administered to maize seeds to protect them against *F. verticillioides* [34]. Previous research has shown that *Trichoderma* spp. can be used to reduce maize stalk rot [35].

In China, seeds are often treated with chemical pesticides before being sold to farmers, making it virtually impossible to re-coat *Trichoderma* on the outer surface of chemically coated seeds. As a result, treating the soil with *Trichoderma* is more practicable. Adopting an environmentally friendly and viable control method is imperative. Over the decades, legumes have established a more significant role in agriculture. Snow pea cultivation has gained a lot of attention; however, root rot is becoming an increasingly important concern. The response of snow peas to pathogen attacks, such as *Fusarium* spp., and their control using bioagents has not well been documented. In addition, there are also fewer studies regarding the use of *Trichoderma* as a biological control agent (BCA) to control *F. solani*- and *F. avenaceum*-induced snow pea root rot. Therefore, the goals of this study were to investigate the mycoparasitic mechanisms of *T. longibrachiatum* (TL6 and TL13) on *F. solani* (FSH) and *F. avenaceum* (FAH), as well as the growth-promoting effect on snow pea seedlings, under *F. solani* and *F. avenaceum* stress.

## 2. Materials and Methods

### 2.1. Fungal Isolates

The present study evaluates the biocontrol activity of two fungal isolates, *T. longibrachiatum* (TL6 and TL13). The biocontrol fungi for this study were acquired from Prof. Li Huixia at Gansu Agricultural University in China. *F. solani* and *F. avenaceum* were isolated from field-infected plants and identified as the pathogens of snow pea root rot in the laboratory.

#### 2.1.1. Medium and Culture

A potato dextrose agar (PDA) medium (potato 400 g, glucose 40 g, and agar 28.2 g) was prepared in the lab and used for the confrontation culture of the FSH and FAH and biocontrol fungal isolates.

#### 2.1.2. Dual-Culture Assay of Antagonistic Fungi and Pathogens

The inhibitory effects of the biocontrol fungi isolate against *F. solani* (FSH) and *F. avenaceum* (FAH) were studied in vitro under modified conditions using a dual-culture-plate method by Rahman et al. [36]. The study was conducted on potato dextrose agar (PDA) media using a colony-diameter-growth technique. The dual-culture method involved parallel transfers of mycelial plugs (5 mm in diameter) from the cultures of FSH and FAH (14 days old) and TL6 and TL13 (3 days old) to the opposite sides of a PDA plate, 1.5 cm from the edge of the plate. The experimental design was completely randomized with only FSH and FAH used as the controls. Each group had six replicates. The plates were incubated at 25 ± 1 °C for 7 days after inoculation. The growth distance between the pathogens (FSH and FAH) in the treatment and the control groups was measured after 7 days. The antifungal effect was then estimated using the formula described by Lu et al. [35] with a slight modification. The inhibition rate was calculated using the following Formula:$$\mathrm{Inhibition}\;\mathrm{rate}\;\left(\%\right)=\frac{\left(\mathrm{Average}\;\mathrm{diameters}\;\mathrm{of}\;\mathrm{the}\;\mathrm{control}-\mathrm{Average}\;\mathrm{diameters}\;\mathrm{of}\;\mathrm{the}\;\mathrm{treatment}\right)}{\left(\mathrm{Average}\;\mathrm{diameters}\;\mathrm{of}\;\mathrm{the}\;\mathrm{control}-0.5\right)}\times 100$$
where the control represents the growth distance of the pathogen colony and treatment represents the growth distance of the pathogens in the treatment dual cultures. Meanwhile, 0.5 represents the diameter of the mycelium plug.

#### 2.1.3. Fungal Inoculum Preparation

Following the dual-culture assay, TL6 and TL13, as well as the pathogenic fungi FSH and FAH, were further grown on the PDA for 7 and 14 days at 25 °C, respectively. TL6 and TL13 conidia suspensions, as well as the pathogenic fungi FSH and FAH, were prepared using the procedure of Zhang et al. [37]. The final suspensions of 10^7^ cfu mL^−1^ were prepared and kept at 4 °C.

### 2.2. Microscopic Observations of the Mycoparasitic Effects of TL6 and TL13 on F. solani and F. avenaceum

A dual-culture-plate test was used to study the physical interactions between TL6 and TL13 with FSH and FAH. The physical interactions of TL6 and TL13 with FSH and FAH were investigated using a dual-culture-plate assay. Following contact (overgrowth) with each other, cellophane fragments were collected from the interaction zone, following the method established by Yassin et al. [38]. A wet-mounted glass slide was prepared, and the interaction zone was imaged using an inverted microscope (Accu Scope, Commack, NY, USA) EXI-410.

### 2.3. Effects of TL6 and TL13 on Snow Pea Seeds Germination under F. solani and F. avenaceum Infections

The snow peas (*Pisum sativum* L. var. *saccharatum*) cultivar ‘L.S. Seed’, provided by Prof. Li Huixia, Gansu Agricultural University (Lanzhou, China), was used in the study.

Snow pea seeds of a relatively uniform size were thoroughly sterilized with 1% (*v*/*v*) NaOCl for 5 min and washed with sterile water 4–8 times. Afterward, snow pea seeds were soaked in a fungi spore suspension (10^7^ spores per mL) and the control was soaked in distilled water (Table 1). According to the method of Zhang et al. [39], seeds were air-dried overnight under aseptic conditions before being sown. The sandy loam soil used in the experiment was obtained from a field of crops in Lanzhou, China (36.061° N, 103.834° E, and 1518 m above sea level). The experiment was set up in a completely randomized design with two controls: plants inoculated with FSH and FAH (positive control) and plants inoculated with sterile water only (negative control). The treatments are detailed in Table 1.

For each treatment, three plastic pots, each containing 200 g of sterile soil measuring 12 cm in diameter and 13.5 cm in depth were used. Ten seeds of equal size were planted 1 cm deep in the soil and gently covered. The plants were watered every 24 h, kept at a constant temperature of 25 °C ± 0.5 with supplemental day/night lighting of 16/8 h, and a relative humidity of 65%. Seedling germination was determined using the method described by Oluwaranti et al. [40], and the percentage of seed germination potential (GP%) was determined as follows:G.P% = {(seedlings germinated after 3 days)/seeds planted} × 100

The germination rate was calculated as [GR (%) = (NGS/TNS) × 100], where NGS is the number of seeds germinated 5 days after planting and TNS is the total number of seeds planted in each pot. The germination index was calculated as [GI (%) = NGSi/Ti × 100], in which NGSi is the number of seeds germinated at a given time, and Ti is the time of incubation, all based on the calculation of Niu et al. [41].

### 2.4. Growth Parameters

The snow pea seedlings were harvested 21 days after treatment. The shoots and roots of the snow pea seedlings were removed, washed three times with distilled water, dried, and weighed. The length and weight of the shoots and roots were determined by the use of a tape measure and a weighing balance. The snow pea shoots and roots were oven-dried at their dried weight at 105 °C for 30 min, and then kept at 80 °C to ensure a steady weight before being weighed. Each preservation and control procedure was carried out three times. According to Tian and Philpot [42], the relative water content (RWC) of the shoots and roots was determined. RWC (%) = [(FW − DW)/FW] × 100, where RWC denotes the relative water content, FW denotes the fresh weight, and DW denotes the dry weight.

### 2.5. Disease Assessment

The disease index of snow pea rot was determined 21 days after treatment. The disease symptoms were characterized using a disease index based on root rot, yellowing, and the chlorosis of cotyledons and leaves at 21 days. The severity of the disease was rated with a 5-degree scale (0–5) described by Zhang et al. [43], where 0 = no disease, 1 = a trace to 10% of the roots were rotted, 2 = 11–25% of the roots were rotted, 3 = 26–50% of the roots were rotted, 4 = 51–75% of the roots were rotted; and 5 = 76–100% of the roots were rotted. The DI was determined using the following Formula:DI (%) = [Σ (number of diseased plants × disease index)/(total number of plants investigated × highest disease index)] × 100

#### 2.5.1. Determination of Chlorophyll Content and Carotenoid 

Chlorophyll was extracted with methanol using the method described by Miazek and Ledakowicz [44]. The fresh leaves of snow pea seedlings weighing 0.2 g were grounded to a powder using liquid nitrogen and homogenized with 10 mL of methanol (BT16R, OLABO Scientific Co., Ltd., Jinan, China). The chlorophyll and carotenoid content were measured using a dual-wavelength spectrophotometer (EPOCH2 Plate Reader, BioTek, Winooski, VT, USA) at 665 and 652 nm absorbance.

#### 2.5.2. Determination of MDA and H_2_O_2_ in Leaves

The levels of malondialdehyde (MDA) and hydrogen peroxide (H_2_O_2_), a product of lipid peroxidation produced by the thiobarbituric acid reaction and an indicator of oxidative damage to a biological system, were measured according to the manufacturer’s protocol/assay kits provided (Solarbio, Beijing, China). The absorbance of the MDA sample was measured at three different wavelengths of 450, 532, and 600 nm, and H_2_O_2_ at 415 nm using a spectrophotometer (EPOCH2 Plate Reader, BioTek, Winooski, VT, USA). The contents of MDA and H_2_O_2_ were expressed as µmol g^−1^ FW [11].

#### 2.5.3. Antioxidant Enzyme Activities in Snow Pea Leaves

The leaf samples were used for antioxidant investigations after 21 days following the snow pea seed treatments. SOD (EC 1.15.1.1), POD (EC 1.11.1.7), and CAT (EC 1.11.1.6) antioxidant activity were evaluated using the assay kits provided, according to the manufacturer’s protocol (Solarbio, China). A spectrophotometer (EPOCH2 Plate Reader, BioTek, Winooski, VT, USA) was used to measure SOD at 560 nm, POD at 470 nm, and CAT at 240 nm. The activities were expressed as U mg^−1^ FW [11].

### 2.6. Statistical Analysis

The data tested in each experiment included TL6 and TL13 strains controlling *F. solani* and *F. avenaceum* and enhancing growth in snow pea seedlings. Data were analyzed using a two-way ANOVA in SPSS Version 16.0 (SPSS Inc., Chicago, IL, USA), mean comparisons were made using Duncan’s new multiple range tests, and the significance was considered at *p* < 0.05.

## 3. Results

### 3.1. In Vitro Colony Growth Inhibition of TL6 and TL13 on F. avenaceum and F. solani

The colony growth of FSH and FAH was significantly (*p <* 0.05) influenced by TL6 and TL13 on various days following incubation. The average inhibitory rates of TL6 against FSH and FAH were 54.58% and 69.16%, respectively, on day 7. Similarly, TL13 against FSH and FAH were 59.06% and 71.27%, respectively, on day 7 (Figure 1B,C,E,F,H,I,K,L).

#### Microscopic Observations of the Mycoparasitic Effects of TL6 and TL13 on *F. solani* and *F. avenaceum*


Mycoparasitic mechanisms, such as the development of appressoria and hook-like structures around the host hyphae of FSH and FAH, enabling firm adhesion to the fungi host, were detected as shown in Figure 2. The growth of TL6 and TL13 conidia in the fungal host was observed, showing the usage of the fungi host as a source of nourishment. Strains TL6 and TL13 demonstrated a mycoparasitic mechanism by penetrating the mycelium of FSH and FAH (Figure 2A,C). Mycoparasitism was revealed by the visible destruction and disintegration of FSH and FAH mycelium by TL6 and TL13, along with the parallel growth in intimate mycelium association. TL13 germinating spore tubes were discovered growing on the mycelium of FSH as shown in Figure 2B. Again, TL6 wrapped around FAH hyphae, thereby degrading the cell wall (Figure 2D).

### 3.2. Effects of TL6 and TL13 on Snow Pea Seeds Germination under F. solani and F. avenaceum Infections

Seed pretreatment with *T. longibrachiatum* (TL6 and TL13) stimulated seed germination (Figure 3). The germination rate (GR), germination potential (GP), and germination index (GI) increased by 8.4%, 5.0%, and 4.3%, respectively, in response to the TL6 treatment compared to the sterile water treatment. However, compared to the control, TL13 increased the GP, GI, and GR by 12.0%, 12.7%, and 17.5%, respectively (Figure 3). In comparison to the control, the *F. solani* treatment of the seeds decreased the GP, GI, and GR by 43.3%, 46.0%, and 48.9%, respectively. Relative to the control, the *F. avenaceum* treatment decreased the GP, GI, and GR by 51.7%, 51.8%, and 27.4%, respectively.

### 3.3. Growth Parameters

The number of leaves, plant height, total fresh weight, dry weight (DW), and relative water content (RWC) of snow pea plants increased significantly at *p* < 0.05 by the TL6 and TL13 treatment as shown in Figure 4. TL6 increased the number of leaves, plant height, total fresh weight, dry weight, and relative water content by 20.7, 9.6, 40.4, 36.6, and 40.7%, respectively, compared to the control. Moreover, TL13 increased the number of leaves, plant height, total fresh weight, and dry weight by 35.4, 18.7, 44.6, 39.8, and 45.4%, respectively, in comparison to the control (Figure 4). The number of leaves, plant height, total fresh weight, and dry weight of TL6-treated plants increased by 16.1, 49.3, 18.4, 56.3, and 23.9%, respectively, compared to the FSH-treated plants. Again, there was an increase in the number of leaves, plant height, total fresh weight, and dry weight by 22.9, 46.6, 22.0, 44.9, and 20.8%, respectively, in the combined TL6 + FAH treatment compared to the control. Furthermore, the combined TL13 + FSH treatment increased the number of leaves, plant height, total fresh weight, dry weight, and relative water content by 29.0, 44.8, 24.6, 60.5, and 27.3%, respectively, compared to the control. The combined TL13 + FAH treatment increased the number of leaves, plant height, total fresh weight, dry weight, and relative water content by 34.4, 40.1, 30.9, 52.6, and 24.1%, respectively, in comparison to control.

### 3.4. Disease Assessment

The findings demonstrated that disease occurrence was extremely severe, as high as 82.3 and 79.7% in the FSH and FAH treatments, respectively, without TL6 and TL13 (Figure 5). However, the administration of TL6 decreased the disease occurrence caused by FSH and FAH by 57.6 and 60.9%, respectively, in comparison to the control. Additionally, TL13 decreased the disease incidence caused by FSH and FAH by 86.6 and 81.6%, respectively (Figure 5 and Figure 6).

#### 3.4.1. Chlorophyll and Carotenoid Contents

Compared to the control, the FSH treatment alone decreased chlorophyll a, b, and the total (a + b) by 58.7, 56.0, and 57.4%, respectively. Similarly, the FAH treatment alone decreased chlorophyll a, b, and the total (a + b) by 54.2, 54.3, and 54.2%, respectively, in comparison to the control as shown in Figure 7A–C. Hence, after administering the TL6 and TL13 treatments, the chlorophyll a, b, and the total chlorophyll content in the stressed snow pea plants caused by FSH and FAH was restored to a level comparable to the control. In addition, when TL6 and TL13 were applied without FSH and FAH stress, the values of chlorophyll a, b, and the total chlorophyll content increased significantly in comparison to the control (Figure 7A–C). Similarly, carotenoid content decreased by 58.9 and 57.7% when treated with FSH and FAH, respectively, in comparison to the control. However, the treatment of TL6 and TL13 increased the carotenoid content in snow pea plants under FSH and FAH stress. The application of TL6 and TL13 without an FSH and FAH infestation significantly increased the carotenoid content in comparison to the control (Figure 7D).

#### 3.4.2. MDA and H_2_O_2_ Accumulation

Snow pea plants treated with FSH and FAH without TL6 and TL13 increased MDA content in leaves by 64.8 and 66.0%, respectively, compared to the control. The MDA content in the combined treatment of TL6 and TL13 with FSH and FAH decreased by 75.6 and 76.8% compared to their respective controls (Figure 8). H_2_O_2_ content increased by 64.4 and 65.9% in FSH- and FAH-treated plants without TL6 and TL13, respectively, in comparison to the control. However, the H_2_O_2_ content in the combined treatment of TL6 and TL13 with FSH and FAH decreased by 70.0 and 76.4%, respectively, in comparison to their respective control (Figure 8).

### 3.5. Antioxidant Enzymes Activities

The stress caused by FSH and FAH in snow pea plants significantly induced and enhanced the antioxidant enzyme activities (SOD, POD, and CAT). SOD, POD, and CAT activities in snow pea plants treated with TL6 increased by 35.2, 56.3, and 43.1%, respectively, in comparison to the control. Similarly, snow pea plants treated with TL13 increased SOD, POD, and CAT activities by 50.2, 69.8, and 52.0%, respectively, compared to the control (Figure 9). However, snow pea plants treated with the combination of TL6 + FSH and TL6 + FAH enhanced the activities of SOD, POD, and CAT by 60.5, 64.7, and 60.3% and 60.0, 64.9, and 56.6%, respectively, compared to the FSH and FAH treatments alone. Again, compared to the FSH and FAH treatments alone, snow pea plants treated with the combination of TL13 + FSH and TL13 + FAH enhanced the activities of SOD, POD, and CAT by 69.7, 68.6, and 65.6% and 70.1, 69.5, and 65.8%, respectively (Figure 9).

## 4. Discussion

*Trichoderma* isolates can suppress fungal phytopathogens directly through mycoparasitism, or indirectly through nutrition and space competition, the promotion of plant development, and by enhancing defense mechanisms. In the current study, *T. longibrachiatum* (TL6 and TL13) grows parallel and tightly wraps around the hyphae of FSH and FAH, showing mycoparasitism [45]. In addition, TL6 and TL13 develop appressoria and hook-like structures that enable solid adhesion to the FSH and FAH hosts. The attachment and germination of TL6, as well as TL13 spores, were detected on FSH and FAH host surfaces, indicating nutrient absorption from the fungi host. These findings are similar to those of Sachdev and Singh [21],who discovered that *Trichoderma* produces appressoria and hook-like structures that enable a solid attachment to the fungal host during mycoparasitism, wrapping around the hyphae of the host. This shows that the host identification and lysis mechanisms used by *Trichoderma* may vary depending on the *Trichoderma*–host interactions. However, in the interaction zone, TL6 and TL13 launched cell-wall-degrading enzyme (CWDE) activity by degrading the hyphae and cell walls of FSH and FAH. These findings were supported by the results of Vinale et al. [46], who discovered that CWDEs may hydrolyze or break down the cell walls of host pathogens and impede their development. 

In the current study, snow pea pretreatment with TL6 and TL13 increased seed germination and plant growth. The effects of the TL6 and TL13 treatments resulted in an increase in plant growth parameters, including shoot height, root length, and the number of leaves. The combined TL6 and TL13 with FSH and FAH also decreased the disease index and reduced root rot and seedlings’ death. Although *Trichoderma* spp. has been shown to promote plant development, there is limited evidence of the systemic responses of plants to *T. longibrachiatum* under FSH and FAH stress conditions. Our results showed that FSH and FAH treatment significantly affected the growth and development of snow pea plants. However, this impact was significantly reduced with the administration of TL6 and TL13. Our findings are similar to several previous studies that found the negative effect of *Fusarium* species on plant seed germination and growth both in vitro and in the greenhouse [47,48]. Again, our findings are similar to previous research that found *T. harzianum* and *T. viride* enhanced seed germination, shoot length, and promoted healthy plant growth [49]. Other researchers have shown that seeds prepared with *T. viride*, *T. harzianum*, and *T. pseudokoningii* inoculant extracts enhanced germination rate and seedling vigor, and decreased the incidence of seed-borne fungi infections in comparison to the control [50,51]. According to the results of the current study, *Trichoderma* species TL6 and TL13 enhanced snow pea seed germination and seedling vigor. However, TL13-treated plants had a significant resistance to FSH and FAH infection compared to TL6 and the control.

In the present study, TL6 and TL13 enhanced the plant height, total fresh weight, dry weight, number of leaves, and relative water content, but TL13 was more effective compared to TL6. However, the FSH and FAH infections decreased the number of leaves, plant height, total fresh weight, dry weight, as well as relative water content. Plants affected by FSH and FAH were often stunted; their leaves turned from pale green to golden yellow and then wilted, withered, died, and gradually dropped from the stem base. The xylem vascular tissue of the roots and the lower stem were dark streaks, and the roots decayed. The FSH and FAH infections increased ROS production in the seedling, leading to low water and nutrient absorption. The abovementioned reasons decreased the photosynthetic activity and led to reduced biomass production in the seedlings. Our results are inconsistent with the findings of Jaiswal et al. [52] in bean, Mayo et al. [53] in tomato and Youssef et al. [54], where the *Rhizoctonia solani* infection was linked to growth suppression in a variety of plants. The growth inhibition caused by the *R. solani* pathogen could be associated with high osmotic stress, plant water-retention disturbance, nutrient depletion, a relatively high respiration rate, reduced photosynthetic activities, and a reduction in the activation of the natural plant-defense mechanisms, resulting in declined total biomass production, as previously reported by Nostar et al. [55], Hashem et al. [56], and Saidimoradi et al. [57]. Several *Trichoderma* species, including *T. harzianum*, *T. longibrachiatum*, and *T. asperellum* under salt stress and *R. solani* infection, have been proven to promote plant development in rice [58], wheat [11,59], bean [53], and cotton [60]. In this study, the snow pea plants’ chlorophyll was influenced by biotic stress. A crucial component of photosynthesis is chlorophyll, and by regulating cellular osmotic processes, plant bio-regulators help enhance chlorophyll quality [61,62]. Under pathogens stress, the reduction in chlorophyll concentration in snow pea plants could be attributed to the breakdown or reduced activities of chlorophyll production enzymes [63,64]. The decrease in chlorophyll content could be linked to an increase in stress symptoms [65]. However, the TL6 and TL13 pretreatment increased the seedlings pigmentation with or without FSH and FAH stress.

Furthermore, MDA and H_2_O_2_ levels increased in the leaves of FSH and FAH -treated plants in this current treatment. However, the TL6 and TL13 treatment reduced cellular damage by decreasing MDA and H_2_O_2_ levels. This was attributable to the increased antioxidant enzyme activities in the snow pea plant by the TL6 and TL13 treatment. The TL6 and TL13 extended their antioxidant enzyme machinery to preserve osmotic balance and metabolic equilibrium in snow pea plants under pathogen stress and enhanced tolerance to oxidative stress. Our findings are consistent with Guler et al. [66] and Shukla et al. [67], who reported that *T. atroviride* and *T. harzianum* inhibited MDA and H_2_O_2_ production in maize and rice roots.

Antioxidant enzymes are proteins that catalyze the conversion of ROS and their byproducts into stable, nontoxic substances, making them the most significant defense against ROS stress [68]. The antioxidant enzymatic mechanism, which comprises the coordinated activities of many enzymes, such as SOD, POD, and CAT, is one of the most essential defensive systems in plants [69]. However, combined TL6 and TL13, with or without FAH and FSH, increased SOD, POD, and CAT activities in snow pea plants, which was consistent with the findings of Boamah at al. [11] where *T. longibrachiatum* TG1 increased SOD, POD, and CAT activity in wheat seedlings under *F. pseudograminearum* stress. An increase in antioxidant enzyme activity decreased lipid peroxidation or ROS production in the seedlings. Similarly, the *T. longibrachiatum* (T6) treatment was reported to increase SOD, CAT, and POD activity in wheat seedlings under salt (NaCl) stress by Zhang et al. [70]. Our results are in agreement with those of Luan et al. [71], who showed that *Trichoderma* isolate ThTrx5 confers salt tolerance to *Arabidopsis* by activating the stress response signals and activities of SOD, POD, and CAT. Chowdappa et al. [72] and Kumar et al. [60] showed that oxidative stress was regulated in infected plants through an increase in antioxidant enzymes, such as POX, CAT, and SOD, in *T. harzianum* and *T. virens*. Similar results was reported in *T. harzianum* by Youssef et al. [54] and in *T. atroviride* by Nawrocka et al. [73]. However, regardless of FAH and FSH infection, TL6 and TL13 increased SOD, POD, and CAT activity in snow pea plants, which is consistent with the findings of Sekmen Cetinel et al. [74], who demonstrated that the role of *T. citrinoviride* in strawberry (*Fragaria* x *ananassa* Duch.) mitigated *Rhizoctonia solani* stress through increased antioxidant activity. Our findings reveal that the coordination of POD, CAT, and SOD activity played a critical protective function in the MDA- and H_2_O_2_-scavenging process in snow pea plants treated with TL6 and TL13. To the best of our knowledge, this is the first time research has been conducted to reveal the involvement of plant-growth-promoting fungi TL6 and TL13 in increasing snow pea plants’ resistance to FAH and FSH infection.

## 5. Conclusions

In conclusion, the results described here show that TL13 and TL6 are both plant-growth-promoting fungi that can control FAH and FSH root rot in snow pea seedlings. TL6 and TL13 decreased FAH and FSH disease incidence in snow pea seedlings. The application of TL6 and TL13, with or without FAH and FSH, increased antioxidant activities in snow pea plants and decreased MDA and H_2_O_2_ contents. The TL13 strain had the greatest performance in terms of pathogen inhibition and snow pea growth promotion. According to the findings, TL13 and TL6 can be employed as biological control agents to protect snow pea seedlings against FAH and FSH in plant disease management.

## Figures and Tables

**Figure 1 jof-08-01148-f001:**
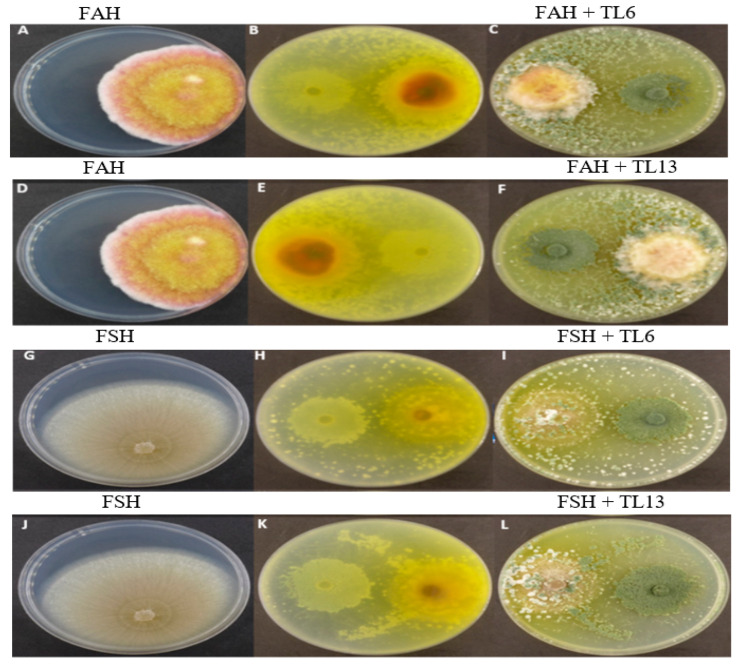
Effects of *T. longibrachiatum* (TL6 and TL13) on *Fusarium avenaceum* (FAH) and *Fusarium solani* (FSH) colony growth and inhibition after 7 days of dual-culture assay. (**A**,**D**) The front colony of *F. avenaceum* cultured for 7 days. (**B**,**C**) The reverse and front colony of FAH and TL6 confrontation culture. (**E**,**F**) The reverse and front colony of FAH and TL13 in confrontation culture. (**G**,**J**) A front colony of *Fusarium solani* (FSH) cultured for 7 days. (**H**,**I**) Reverse and front colony of FSH and TL6 confrontation culture. (**K**,**L**) FSH and TL13 confrontation culture of reverse and front colony.

**Figure 2 jof-08-01148-f002:**
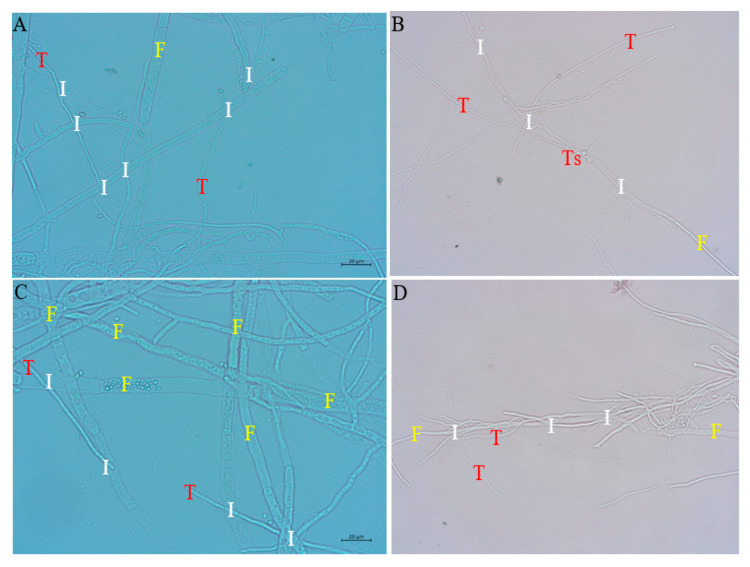
Mycoparasitism of *T. longibrachiatum* (TL6 and TL13) against *Fusarium solani* (FSH) and *Fusarium avenaceum* (FAH) was observed under light microscopy at ×40 magnification. (**A**) Internal and external growth of TL6 conidial on *F. avenaceum* hyphae. (**B**) TL13 spore germ tube (Ts) developing on the hyphae, with TL13 conidial growing parallel to *F. solani*, demonstrating hyphae depression. (**C**) TL13 penetration, disruption, and breakage of *F. avenaceum* hyphae and (**D**) TL6-wrapped around *F. solani* hyphae, causing cell wall degradation. T represents *Trichoderma* hyphae, F represents *Fusarium* hyphae, and I represents the point of interaction.

**Figure 3 jof-08-01148-f003:**
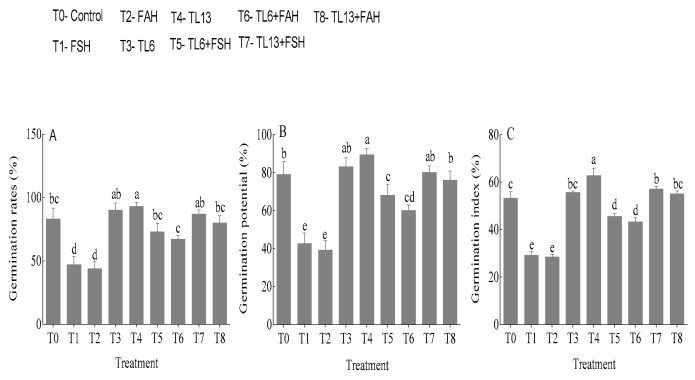
Effects of *T. longibrachiatum* (TL6 and TL3) on snow pea seeds’ (**A**) germination rate, (**B**) germination potential, and (**C**) germination index for *F. solani* (FSH) and *F. avenaceum*. Data are presented as mean ± standard errors (SE) of two independent experiments in three replicates. Means with the same lowercase letters are not significantly different at *p* < 0.05, according to Duncan’s multiple range tests.

**Figure 4 jof-08-01148-f004:**
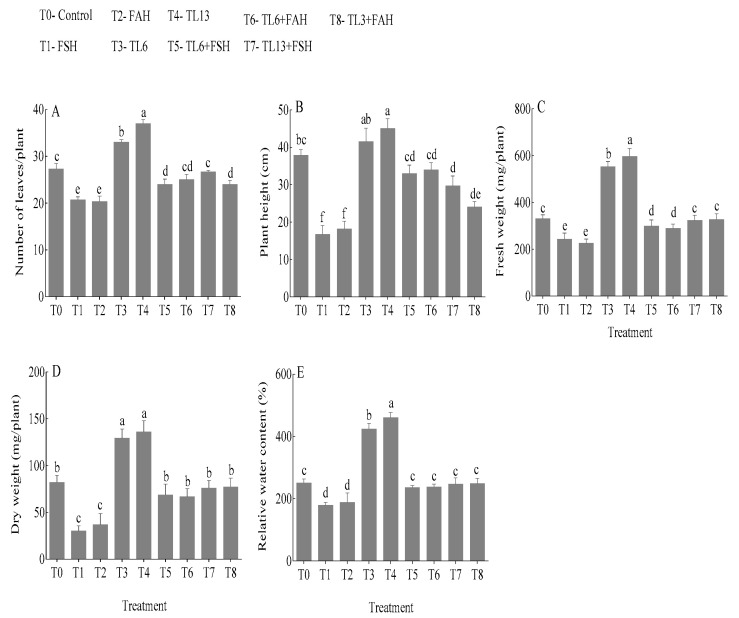
Effects of *T. longibrachiatum* (TL6 and TL3) on snow pea seedling biomass and relative water content at 21 days after pretreatment with *F. solani* (FSH) and *F. avenaceum* (FAH). (**A**) Number of leaves per plant; (**B**) plant height (cm); (**C**) leaves’ fresh weight (mg/plant); (**D**) leaves’ dry weight (mg/plant); (**E**) relative water content (%). Data are presented as mean ± standard errors (SE) of two independent experiments in three replicates. Means with the same lowercase letters are not significantly different at *p* < 0.05, according to Duncan’s multiple range tests.

**Figure 5 jof-08-01148-f005:**
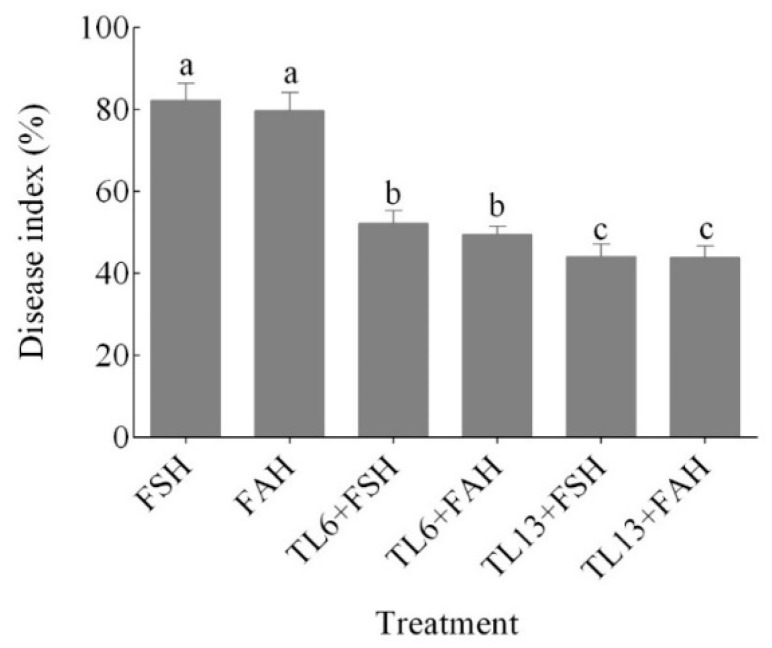
Effects of *T. longibrachiatum* on *F. solani* (FSH) and *F. avenaceum* (FAH) disease index in potted snow pea plants. Data are presented as mean ± standard errors (SE) of two independent experiments in three replicates. Means with the same lowercase letters are not significantly different at *p* < 0.05, according to Duncan’s multiple range tests.

**Figure 6 jof-08-01148-f006:**
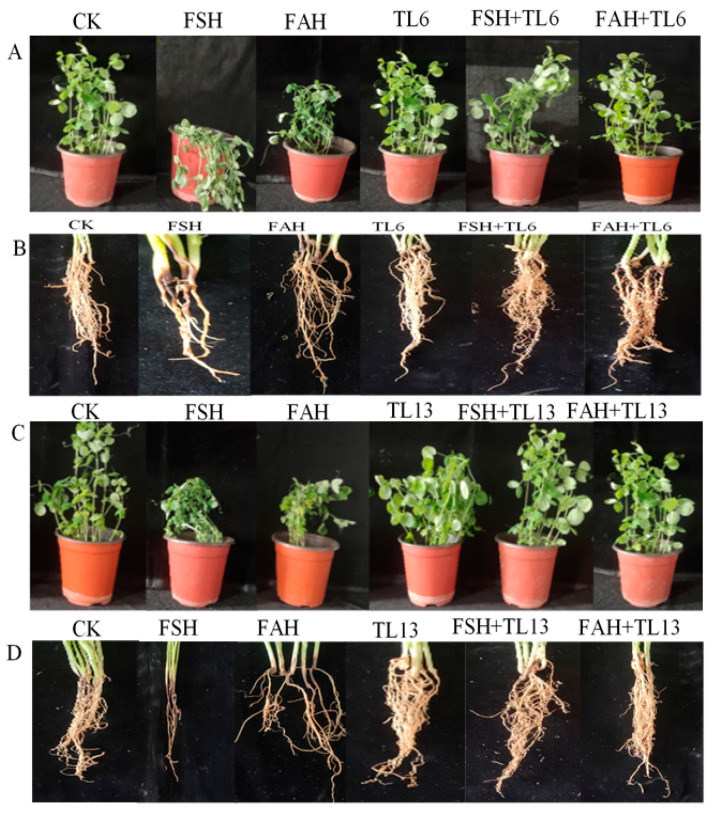
Effects of *T. longibrachiatum* (TL6 and TL3) on snow pea seedling disease index at 21 days after pretreatment. Panels (**A**,**C**) stems and leaves; and (**B**,**D**) roots, showing incidence of disease on snow pea plants after treatment.

**Figure 7 jof-08-01148-f007:**
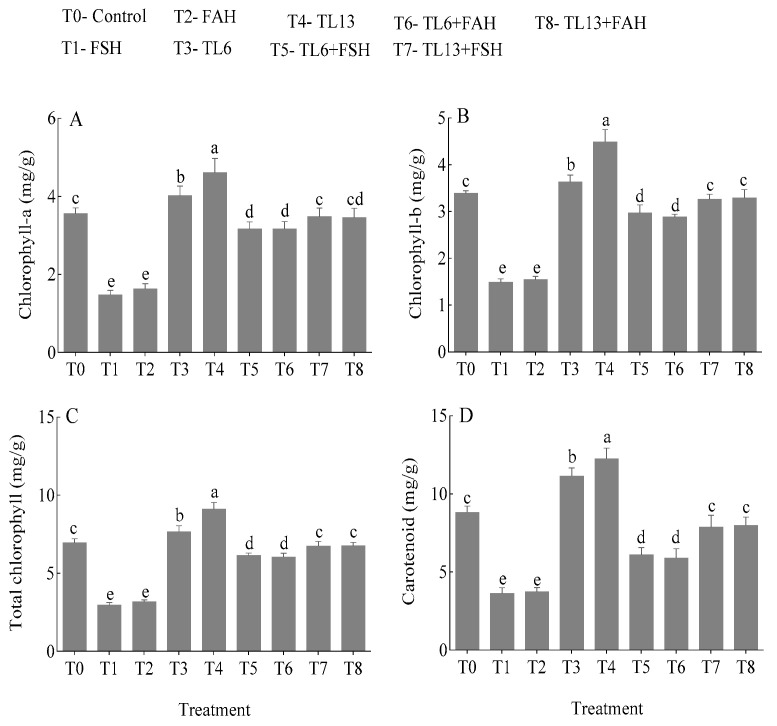
Effects of TL6 and TL13 treatment on (**A**) chlorophyll a, (**B**) chlorophyll b, (**C**) total chlorophyll, and (**D**) carotenoid content in the leaves of snow pea plants under *F. solani* (FSH) and *F. avenaceum* (FAH) stress. Data are presented as mean ± standard errors (SE) of two independent experiments in three replicates. Means with the same lowercase letters are not significantly different at *p* < 0.05, according to Duncan’s multiple range tests.

**Figure 8 jof-08-01148-f008:**
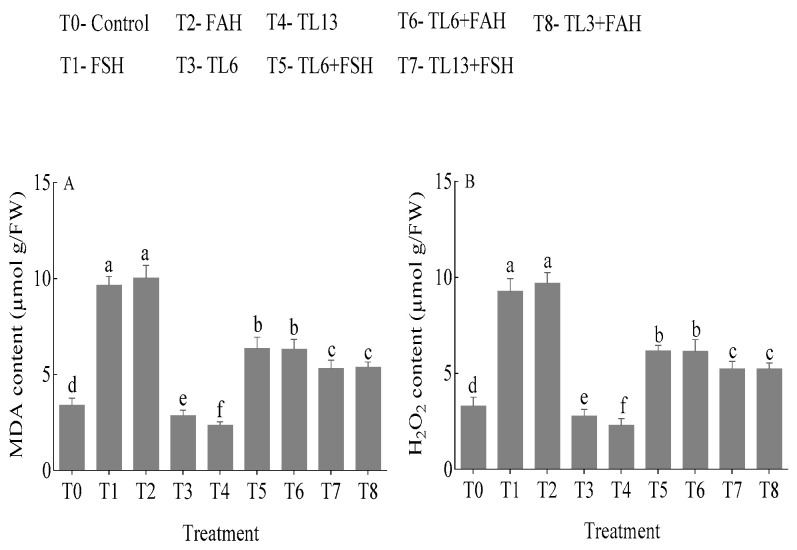
Effects of TL6 and TL13 treatment on (**A**) MDA and (**B**) H_2_O_2_ in the leaves of snow pea plants under *F. solani* and *F. avenaceum* stress. Data are presented as mean ± standard errors (SE) of two independent experiments in three replicates. Means with the same lowercase letters are not significantly different at *p* < 0.05, according to Duncan’s multiple range tests.

**Figure 9 jof-08-01148-f009:**
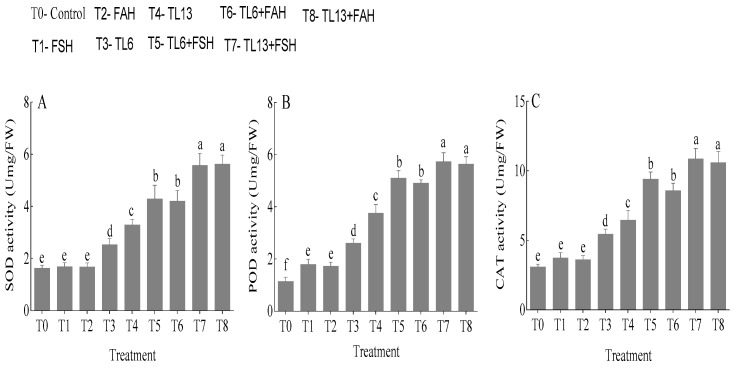
Effects of TL6 and TL13 treatments on (**A**) SOD, (**B**) POD, and (**C**) CAT activities in the leaves of snow pea plants under *F. solani* and *F. avenaceum* stress at 21 days after pretreatment. Data are presented as mean ± standard errors (SE) of two independent experiments in three replicates. Means with the same lowercase letters are not significantly different at *p* < 0.05, according to Duncan’s multiple range tests.

**Table 1 jof-08-01148-t001:** Description of treatments.

Group	Name	Treatment
T0	Negative control	Distilled water
T1	Positive control	*Fusarium solani* (FSH)
T2	Positive control	*Fusarium avenaceum* (FAH)
T3	TL6	*Trichoderma longibrachiatum* (TL6)
T4	TL13	*Trichoderma longibrachiatum* TL13
T5	TL6 + FSH	Inoculation of strain TL6 + FSH
T6	TL6 + FAH	Inoculation of strain TL6 + FAH
T7	TL13 + FSH	Inoculation of strain TL13 + FSH
T8	TL13 + FAH	Inoculation of strain TL13 + FAH

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
