# Peer review of "Antagonistic Effect of Trichoderma longibrachiatum (TL6 and TL13) on Fusarium solani and Fusarium avenaceum Causing Root Rot on Snow Pea Plants"

_jof, 2022, doi:10.3390/jof8111148_

Round 1

Reviewer 1 Report

Species from the genus Trichoderma are  the world's most known antagonist with various mechanisms of action, including mycoparasitism, due to the production of numerous toxins and enzymes. They also enhance the plant's defense mechanisms. The present paper provides very valuable data documenting the biocontrol role of Trichoderma longibrachiatum at various levels. These are very valuable results due to the great importance of Pisum sativum in agriculture and the need to limit the use of chemical fungicides. Overall, the experiments are numerous, very interesting and well done. The results are presented in detail and in a clear form. The discussion is interesting. However, there is a lot of mistakes or missing parts in the whole manuscript. Even in the title there is a mistake in the name of the fungus. For one pathogen, three different names are given in the text: Fusarium avenaceum (correct), F. avenacaeum (error) or F. aveaneum (error). After considering the comments below and rereading the entire text, the manuscript deserves to be published in the Journal of Fungi.

Title Line 3 is the wrong fungus name, it should be Fusarium avenaceum

Line 27   MDA  - explanation  is only written in line 215, it should be explained here

Line 42 Snow peas (Pisum sativum), the Latin name should be clarified, is it may be some varietas?, 'Pisum sativum' is in English 'common pea'

Line 74, these indicators require an explanation here of SOD, POD and CAT, although it is given in the abstract

Line 113 it is Fusaria      it should be Fusarium spp. / or Fusarium species

Line 117 it is F. solani (FSH) and F. avenacaeum,     it should be consistently given 'F. solani (FSH) and F. avenacaeum (FAH)'

Line 122 it is ′ F. Solani and F. Avenaceum  it should be F. solani and F. avenaceum

Line 125 it is' F. Solani      it should be ´F. solani

Line 124 Fungal isolates – it only says '…. acquired from Prof. Li Huixia at Gansu Agricultural University in P. R. China' it is far too little in the present decade.  Reference should be made to the paper if it is published, if not now state whether the identification was only on a morphological basis or molecularly as well. At least ITS should be done, and GenBank acession number given. Proper identification of this species is very important, as this is what all of the present work is based on.

Line 132 it is' F. Solani ´ it should be ´ F. solani ’. Correction must be made throughout the manuscript - the fungal species name must be lowercase

Line 129 - (PDA) medium - to be specified by the manufacturer

Line 163 - Pisum sativum - it should be italic. In addition, the entire manuscript should be corrected so that Latin names of plant and fungi are spelled in italic (eg Line 243,379, 399,410, 433 ...)

Line 167 – ml, line 150 - mL - the record should be standardized

Line 179 required correction of '[35]method'

Table 1 why is the symbol given only  for F. solani (FSH) and there is no ‘(FAH)’ for F. avenaceum? Does it have a purpose?

Line 214 H2O2 needs correction

Line 236 it is F. avenacaeum it should be F. avenaceum

Line 237 it is F. avenacaeum it should be F. avenaceum

Line 255 and line 266, 270, 277, 303, 313, 330, 340, 344, 348, 358, 364, 366,368, 380, 400, 418, 464, 468, 473 it is F. avenacaeum it should be F. avenaceum

Line 252 should be moved to line 253 from region Figure 1 to the text !!

Line 264, line 281, line 390 it is F. avenaeum it should be F. avenaceum

Figure 1 Line 251 there is an error in the data for Figure 1

Figure 2  A, B, C are missing in the description? the description is currently unclear

Line 277 whether the phrase is correct: what does "the F. avenaceum therapy" mean?

Line 280 why Germination and not germination

Figure 4 the explanation  requires correction

Figure 6 Line 322 the parenthesis should be removed

Figure 6 Explanation  to be completed, panel A, C-stems and leaves, B, D -roots?

Figure 8 TL6 and TL13 treatment, which treatment is TL6, which TL13

Line 363 Is the citation of Figure 7 correct? rather, it should be Figure 9

Figure 9 what is A, B, C ???

Line 384 '[15]where'   it should be corrected

Line 389 CWDE - needs explaining

Line 405 it is T. viridi     it should be T. viride

Line 426-427 errors should be corrected

Line 444  needs correction

Line 449, 452, 454 H2O2 needs correction

Line 461 it is F. pseudograminearium    it should be F. pseudograminearum (see Index fungorum)

Line 504 phaseoli - it should be italic, also other names in the Literature

Line 507 numerous errors

Line 522 Mycorrhizal not italic

Line622 Oryza sativa - in italic

Line 577 J JPPM ??

Line 580   f.  it should be not italic

Line 594 it is Pseudoperospora cubensis    it should be Pseudoperonospora cubensis

Line 598 it is Fragaria xananassa   it should be Fragaria x ananassa

Line 600 it is Trichoderma harzmnum it should be Trichoderma harzianum

Line 607 J Planta - it should be Planta, there are also mistakes in the names of other journals, the symbol J was often added unnecessarily

Line 613 Correa - not italic

Line 625 - no journal name

Author Response

Dear Editor, please see the attachment below containing the responses of the comments.

Reviewer 2 Report

  1. I will suggest authors to add only major findings in the abstract part. Make it short and effective.
  2. Did the authors tested the compatibility of two fungal isolates, if so, add the details.
  3. I will suggest authors to add details in the figure legends such as the type of statistical tests performed as well as about error bars (standard deviation or error). What is each alphabets denoting on the bars?
  4. Elaborate antioxidant production in the discussion part.  
  5. Please see the attached file for more corrections.

Author Response

(The authors gave the same response as above.)

Reviewer 3 Report

This study examines the efficacy of Trichoderma as a biocontrol agent against Fusarium solani and F. avenacaeum. The ms was well organized, however, authors need to explain in detail why chlorophyll content, carotenoid, MDA, H2O2, antioxidant enzyme activities should be determined in Introduction. In line 71-82, authors mentioned about reaction of plants against stress, but why authors should analyze in the Trichoderma-Fusarium-snow pea system? Without these analyses, authors appears to achieve the goals of this study.

The ms included some minor errors such as lowercase and uppercase letters.

Author Response

(The authors gave the same response as above.)

Reviewer 4 Report

Dear Authors

I recommend that this paper be rejected for two reasons.

1. Language, writing and spelling issues, even in the title species name mentioned Fusarium avenacaeum [avenaceum] incorrectly. Also, the manuscript is written with poor language quality, and readers cannot understand the current format. 

2. Scientific content: Week research motivation and results not precise, poor analysis and inappropriate statistical data.

Author Response

(The authors gave the same response as above.)

Round 2

Reviewer 4 Report

Dear Author,

In recent, biocontrol agents through beneficial microbes have gained much attention. Trichoderma is the most studied fungal biocontrol agent and control pest, and it may be a promising candidate for future sustainable agriculture. However, the present format of the article is of great interest to researchers in agriculture development. So, the manuscript can be recommended for publication in the Journal of Fungi after minor revisions.

Queries: Line 63: “Under water-stress conditions, root rots are very severe” [Abeysinghe, 2012]; reframe the sentence. 

Line no 215: Where do you keep the experimental pots? An open environment is an else closed condition (Greenhouse). The authors should have revealed the statement in the manuscript materials method section. 

Snow peas [Fabaceae] host beneficial bacteria (rhizobium) associated with the plant, If any adverse effect on the bacteria by the Trichoderma species?